# Tuning LLM Judge Design Decisions for 1/1000 of the Cost

**David Salinas** [1 2]  **Omar Swelam** [1]  **Frank Hutter** [1 2]

## Abstract

Evaluating Large Language Models (LLMs) often requires costly human annotations. To address this, LLM-based judges have been proposed, which compare the outputs of two LLMs enabling the ranking of models without human intervention. While several approaches have been proposed, many confounding factors are present between different papers. For instance the model, the prompt and other hyperparameters are typically changed at the same time making apple-to-apple comparisons challenging. In this paper, we propose to systematically analyze and tune the hyperparameters of LLM judges. To alleviate the high cost of evaluating a judge, we propose to leverage multi-objective multi-fidelity which allows to find judges that trade accuracy for cost and also significantly reduce the cost of the search. Our method identifies judges that not only outperform existing benchmarks in accuracy and cost-efficiency but also utilize open-weight models, ensuring greater accessibility and reproducibility. The code to reproduce our experiments is available at this repository https://github.com/geoalgo/judgetuning.

## 1. Introduction

Instruction tuned models are difficult to evaluate as they provide free-form text given arbitrary instructions that may include summarization (Zhang et al., 2024), code writing (Ni et al., 2023) or translation (Elshin et al., 2024). While humans can annotate the quality of the outputs of an LLM given instructions, this quickly becomes expensive and also delays the evaluation and development of instruction-tuned models (Li et al., 2023).

As an alternative, LLM judges have been proposed to provide an indicative ranking for instruction-tuned models (Li et al., 2023), but also to select instruction-tuning recipes (Grattafiori et al., 2024; Lambert et al., 2024). While they can reduce cost significantly compared to human evals, LLM judges have limitations as they may rely on superficial style aspects, such as the length of a response (Dubois et al., 2024) or the order of their input (Zheng et al., 2023).

While some of these issues can be easily fixed, several aspects limit the improvement of LLM judges. The first is that evaluating a LLM judge configuration is expensive. For instance, evaluating one model in Alpaca-Eval (Li et al., 2023) costs ∼24$ (Ni et al., 2024), and evaluating a *judge* multiplies this by the large number of model evaluations it has to do in order to compute the correlation with human ratings. Given this, many confounding factors are typically present between different judges approaches. For instance, between Alpaca-Eval (Li et al., 2023) and Arena-Hard (Li et al., 2024), the judge model, the prompt, the score-type and the set of instructions were changed. This makes it hard to learn which contributions are important.

In this work, we propose to analyze and tune systematically the hyperparameters of an LLM judge, including the LLM model used, the prompt, the inference parameters (such as the temperature) as well as the parsing mechanism used to extract the judge preference. We first analyze the impact of scaling the LLM judge model or the number of instructions on the judge performance. This highlights that scaling is insufficient to reach good performance and also helps us to identify a cheaper way to evaluate judges than evaluating Spearman correlation with Chatbot Arena with a grid of model annotations. We then show how to systematically tune the hyperparameters of a judge using a multi-objective (to account for accuracy and cost) and multi-fidelity approach which saves tuning cost by stopping early poor configurations. Finally, we show that the configurations found outperform previous state-of-the-art judges on a range of real-world test datasets.

Our key contributions are the following:

- We study scaling laws of judges showing how much model size and number of instructions alone impact key metrics used to evaluate judges

- We propose a way to tune judges hyperparameters at reasonable tuning cost

---

[1]University of Freiburg [2]ELLIS Institute Tübingen. Correspondence to: David Salinas <david.salinas.pro@gmail.com>.

*Proceedings of the 42nd International Conference on Machine Learning*, Vancouver, Canada. PMLR 267, 2025. Copyright 2025 by the author(s).

- We show the approach is able to identify configurations that outperform previous approaches while relying solely on open-weight models

- We analyze which prompt strategy and hyperparameters work best for judges highlighting important patterns that may be used by the community to build better judges

## 2. Related work

**LLM judges.** LLM as a judge has been emerging as a way to alleviate large human annotation costs that are required to evaluate instruction-tuned models. For instance, Llama3 used an earlier version of itself as a reject sampler to select best completions (Grattafiori et al., 2024) or more recently (Lambert et al., 2024) used an LLM judge to annotate preference data to perform instruction tuning. LLM judges have also been used for leaderboards such as Alpaca-Eval (Li et al., 2023) or Arena-Hard (Li et al., 2024) which offer cheaper alternatives than human-annotated leaderboards such as ChatBot Arena (Chiang et al., 2024).

**Zero-shot and fine-tuned judges.** Two main approaches have been proposed for LLM judges. The first one, referred to as zero-shot judges, prompts a LLM to rate a pair of model completions (Li et al., 2023; 2024) or a single completion possibly against a baseline (Zheng et al., 2023). The second one fine-tunes an existing model on a set of human-annotated preferences (Zhu et al., 2023; Wang et al., 2024). In contrast with the first approach, it requires fine-tuning a model which may occur at an extra cost, along with lower robustness under distribution shift (Huang et al., 2024).

**Zero-shot judges.** Many strategies for zero-shot judges have been proposed. Li et al. (2024) proposed to ask the judge to answer the instruction to perform some form of Chain of Thought (Wei et al., 2022). To avoid the order of outputs to matter in pairwise comparison, previous work proposed to randomize or average the two possible positions (Dubois et al., 2024; Li et al., 2024). To parse the model output, Li et al. (2023) asks the judge a letter to denote the best model, Li et al. (2024) uses instead a Likert scale (such as A»B, A>B, A=B to indicate respectively when model A is much better, better or comparable wrt B), and another alternative (Cui et al., 2024; Lambert et al., 2024) outputs a score for various criteria such as instruction-following, honesty, or helpfulness. The underlying LLM models often vary between papers, along with multiple other dimensions, making it difficult to determine which strategies inherently perform better.

**Judge limitations.** In parallel with their adoption, several limitations of LLM judges have been highlighted. Among them is their dependence on superficial stylistic aspects of an answer, for instance, favoring longer answers (Dubois et al., 2024), the first answer when using judges that make pairwise comparison (Li et al., 2024), or their own outputs (Panickssery et al., 2024).

While some of those issues have been fixed by averaging the order of the answers (Li et al., 2024) or using a causal model to isolate the length effect from the judge preference (Dubois et al., 2024), the challenge of favoring its own answer is more problematic. Such biases pose some challenge in aligning with human evaluations which requires more intricate design for the scoring methods (Liu et al., 2024). Indeed, many previous works used judges from close models such as GPT-4 which may bias the leaderboard or render evaluations infeasible if the model becomes unavailable or its cost increases.

**Prompt optimization.** LLM judges tend to be very reliant on the design on optimization of the prompts used. Such strong sensitivity makes the prompt design an important aspect addressed in previous works. Zhou et al. (Zhou et al., 2024) introduce a prompt optimization framework for addressing such sensitivity even for semantically equivalent instructions. Other works (Shi et al., 2024) have explored the importance of prompt optimization at a constrained budget which is essential for scaling. Another important factor is how prompt tuning, together with fine-tuning, can lead to boosted performance (Soylu et al., 2024).

**Judge tuning.** To the best of our knowledge, hyperparameter tuning for LLM judges has not been comprehensively explored in a way that accounts for the various factors contributing to a high-performing judge. One reason is the associated compute cost with naive approaches. For instance, Alpaca-Eval and Arena-Hard evaluate each judge configuration across a grid of models and instructions, leading to substantial expenses when comparing multiple judges[1]. Cost is therefore a strong limitation for the tuning of judges limiting the comparison and tuning of judges to simple aspects such as the choice of the underlying LLM model but excluding other key factors such as the prompt strategy, the output format or the temperature.

In this work, we propose a method to search for optimal judge configurations, including the best prompt parameterization. While prompt tuning approaches such as (Fernando et al., 2023) are related, they do not specifically address tuning judges. A key distinction is that we focus not only on optimizing prompts but also on other judge hyperparameters, such as model selection and temperature. Similarly,

---

[1]We estimate that our approach costs approximately 2K$ to search through 4,480 judge configurations and that evaluating the same number of judges using Alpaca-Eval or Arena-Hard methodology would cost around 2M$, see B.3 for details.

Doddapaneni et al. (2024) analyzed the effectiveness of five different prompt strategies for LLM judges. However, their work primarily introduces a benchmark, whereas our approach is centered on systematically tuning and analyzing judges within a search space that includes 80 prompting strategies and a total of 4,480 judge configurations.

We begin by providing background on LLM judges and examining the extent to which improvements can be achieved through scaling alone. This analysis underscores the importance of human agreement as a more efficient metric for differentiating between judges. We then demonstrate how judges can be effectively tuned by optimizing their prompts and other hyperparameters, such as model selection and temperature.

## 3. Background

### 3.1. Judge

We denote a LLM model as a function $\pi : p \rightarrow o$ that produces an output string $o$ given a prompt $p$. A judge compares two models $\pi_0$ and $\pi_1$ and outputs a score $\Phi(p, \pi_0(p), \pi_1(p)) \in [0, 1]$ which is close to zero if $\pi_0(p)$ is better than $\pi_1(p)$ or close to one otherwise.

A common choice is to use a fixed baseline $\pi^*$ for one of the models, typically a frontier model which allows to obtain a score for a model to be evaluated: $\Phi(p, \pi(p)) = \Phi(p, \pi^*(p), \pi(p))$. Given that the order can sometimes influence the judge decision (Li et al., 2023), previous works have proposed to either randomize the baseline position (Li et al., 2023) or compute the judgement given both orders and average out the result (Li et al., 2024).

### 3.2. Judge metrics.

**Spearman correlation.** One approach to see how well a judge performs is to check how close are its rankings on a grid of models and instructions compared to rankings obtained through human annotations.

Assume we are considering a finite set of models $\mathcal{M}$ and instructions $\mathcal{P}$ and that we are given a list of golden scores for the models denoted $s^h \in \mathbb{R}^{\mathcal{M}}$, for instance, the elo-ratings from the Chatbot Arena obtained from human ratings.

To obtain scores from the judge, we average for each model the preference against a baseline $\pi^*(p)$, e.g.

$$s_i^\Phi = \mathbb{E}_{p \sim \mathcal{P}} \left[ \Phi(p, \pi^*(p), \pi_i(p)) \right].$$

One can then use the Spearman correlation between the scores estimated with the judge and the golden score, e.g. $\rho(s^h, s^\Phi) = \frac{\text{cov}\left[ \text{R}(s^h), \text{R}(s^\Phi) \right]}{\sigma(\text{R}(s^h))\sigma(\text{R}(s^\Phi))} \in [-1, 1]$ where $\text{R}(x)$ and $\sigma(x)$ denotes respectively the rank operator and the standard deviation. Using Spearman correlation is beneficial because

it accounts for differences in scale between the golden scores and the judge annotations, focusing on the ranking consistency rather than absolute values. Other metrics which can be used to compare the order of models include Brier score and calibration described (Li et al., 2024).

**Human agreement.** Another approach to evaluating judges is to measure their agreement with human-annotated preferences in a list of pairwise model battles. Each battle consists of a prompt, a pair of model outputs, and a binary label indicating which output the human annotator preferred.

Let us denote a set of annotated battles:

$$(p_i, o_i, o_i', \Phi^h(p_i, o_i, o_i'))_{i=1}^N \tag{1}$$

where $o_i, o_i'$ denotes the output of two models from prompt $p_i$ and $\Phi^h(p_i, o_i, o_i') \in \{0, 0.5, 1\}$ denotes a human choice for respectively $o_i$, a tie or $o_i'$.

To evaluate the judge quality, one can measure the human agreement which is the percentage of times where the human and judge agrees, e.g.

$$\mathbb{E}_{i \sim N}[\Phi(p_i, o_i, o_i') = \Phi^h(p_i, o_i, o_i'))] \tag{2}$$

Having defined two metrics to evaluate judges, we next evaluate how those are impacted by scaling the base models or number of instructions.

## 4. Scaling judges

Rather than tuning judges a natural question is: can we just scale them up? LLM judges can be scaled in multiple ways: by increasing the number of parameters of the LLM judge or by using more instructions. Here we investigate how well the Spearman correlation and human agreement metrics scale with both dimensions.

**Spearman correlation.** In Fig. 1, we show the scaling behavior when increasing the LLM judge size and the number of instructions. We use Qwen2.5 with a default prompt and compute Spearman correlation on a common set of 26 models available on both Alpaca-Eval and Arena-Hard.

As expected, the judge performance improves when scaling the LLM model and when using more instructions as it allows to cover more areas of the models to evaluate. Alpaca-Eval contains easier prompts and therefore gives better performance to smaller judges compared to Arena-Hard. In contrast, Arena-Hard contains harder and technical questions. This allows to achieve better Spearman correlation as the instructions allow better separability among advanced models, provided that the judge base model is strong enough to measure the performance on this more complex set of instructions.

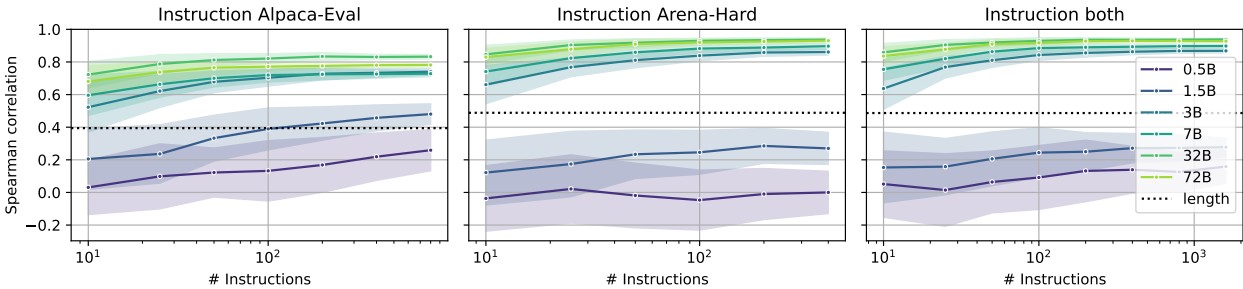

*Figure 1.* Effect of scaling the LLM judge and increasing the number of instructions on Spearman correlation. In contrast to human agreement, neither Alpaca-Eval, Arena-Hard, nor their union distinguishes the quality difference between 32B and 72B models.

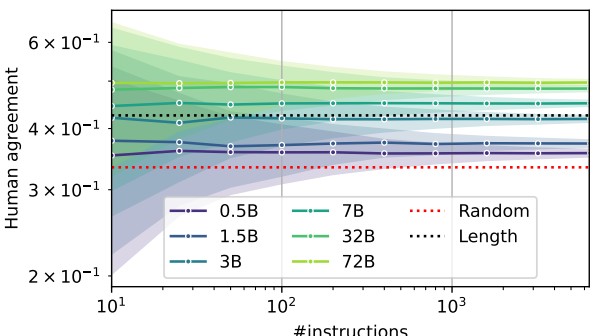

*Figure 2.* Effect on scaling the LLM judge and the number of instructions on human-agreement.

| $\#_{\text{params}}$ | Sp. corr. ($\uparrow$) | CV($\downarrow$) | Hum. agr. ($\uparrow$) | CV($\downarrow$) |
|---|---|---|---|---|
| 0.5B | $0.09 \pm 0.189$ | 207.60 | $0.36 \pm 0.006$ | 1.70 |
| 1.5B | $0.33 \pm 0.137$ | 41.26 | $0.37 \pm 0.006$ | 1.61 |
| 3B | $0.82 \pm 0.066$ | 8.11 | $0.42 \pm 0.006$ | 1.45 |
| 7B | $0.83 \pm 0.082$ | 9.84 | $0.45 \pm 0.006$ | 1.33 |
| 32B | $0.90 \pm 0.052$ | 5.75 | $0.48 \pm 0.006$ | 1.29 |
| 72B | $0.88 \pm 0.074$ | 8.43 | $0.50 \pm 0.006$ | 1.27 |

*Table 1.* Comparison of the variability of Spearman-correlation and Human-agreement metrics when using 6500 random annotations and the same default prompt for all model sizes.

**Human agreement.** In Fig. 2, we study the effect of scaling the LLM judge and the number of battles this time on human agreement. Here, we use the same prompt and base LLM models as in the previous paragraph but compute human agreement on the LMSys dataset (lin Chiang et al., 2024).

We also observe that scaling the LLM judge base model improves performance. However, compared to the previous case, the performance is stationary which is expected since human-agreement is the average of an instruction based property. In contrast, Spearman correlation gets better with more instructions as shown in Fig. 1 as the judge gets a better sense of a model performance (more scenarios are seen and with larger frequency).

**Which metric to optimize.** We have two metrics to evaluate the judge quality: the Spearman correlation and the human agreement. As discussed in the previous paragraph, they behave differently as human agreement is an average of an atomic property (does the judge and human agrees on instruction on average) whereas Spearman correlation is a global metric requiring evaluating many models to compare

rankings. Both metrics are correlated, we can see for instance that the ranking w.r.t. the base model size used for the judge is mostly consistent across both Spearman correlation and human agreement in Figs. 1 and 2. For both metrics, we observe that, for this prompt and model family, LLM judges bellow 7B are not able to outperform a simple length baseline which favors the output with the longest answer.

In Table 1, we compare the variability of both metrics when using a random set of 6500 instructions[2]. We report the standard deviation and Coefficient of Variation (CV) which is computed as $\sigma/\mu * 100$ and indicates the percentage of variation of a metric. It can be seen that while both metrics are correlated, human-agreement allows to differentiate larger models much more easily as the signal to noise ratio of the metric is better. In particular, it allows to statistically distinguish 32B and 72B models whereas those models are tied for Spearman correlation given the number of instructions considered.

In what follows, we therefore use human agreement as the metric to optimize as it allows to rank judge configurations much more cheaply than Spearman correlation and allows to separate judge configurations with much fewer instructions.

---

[2]In the case of Spearman correlation, we sample 250 instructions from both Alpaca-Eval and Arena-Hard datasets (which yields to 6500 annotations given that we have 26 models). For LMSys, we just sample 6500 random instructions.

# 5. Tuning judges

We first describe the search space used - also summarized in Table 5 - which includes searching for the inference and prompt hyperparameters.

## 5.1. Inference hyperparameters

For the LLM model, we search among 7 open-weights options: Llama3.1 (8B and 70B), Qwen2.5 (7B, 27B and 70B) and Gemma2 (9B, 27B). All models with more than 9B parameters are quantized with half-precision. We also search for the LLM temperature in [0.0, 0.01, 0.1, 1.0] and whether to average predictions when considering two possible orders or using just a single order.

## 5.2. Prompt hyperparameters

We now describe how we parametrize different prompt options and we illustrate one such option in Fig. 3.

**Output format.** When prompting a LLM judge, we must be able to parse its output into a preference. We consider the following options where the judge outputs:

- *best-model-identifier*: a letter corresponding to the best assistant as in Li et al. (2023)

- *likert*: a likert scale (such as A»B, A>B, A=B to indicate respectively when model A is much better, better or comparable wrt B) as proposed in Li et al. (2024)

- *pair*: a score for both assistants in [0-10] similar to (Zhu et al., 2023)

- *preference*: a score in $[0, 1]$ where 0 (resp. 1) indicates a preference for model A (resp. B)

- *multi*: the average score for 5 criteria - conciseness, clarity, adherence, comprehensiveness and style similar to Cui et al. (2024); Lambert et al. (2024).

**Provide answer or other information.** Asking a LLM to reflect before providing its answer is known to be beneficial in cases requiring reasoning (Wei et al., 2022). In addition, providing example (few-shot learning) can also be beneficial as shown in (Zheng et al., 2023). We therefore search for the following options and ask the judge to provide before its preference:

- *confidence*: its confidence on its preference

- *answer*: its own answer to the instruction as proposed in (Li et al., 2024)

- *explanation*: its explanation on the given preference

Providing its confidence is meant to help the judge to provide more calibrated preference scores (e.g. to not give a strong score for one option when it is uncertain) while the latter two are meant to elicit chain of thought reasoning.

To use one of the three options, we add in the prompt a description of the option and asks the LLM to generate it, see Fig. 3 for an illustration where the prompt asks the judge to provide its answer and an explanation on its judgement.

**JSON formating.** Formats used to query the output of an LLM have different trade-offs (He et al., 2024). Some are more controllable such as JSON but may loose performance against simpler format (such as raw text) in particular given less capable models.

We search for two formats, using either JSON or raw text. When using JSON, we generate the prompt so that the template asks the LLM to provide a JSON with all the fields needed (the preference in the right format, how to provide its explanation/answer/confidence when needed). In addition, we enforce the LLM to provide a valid JSON by only selecting completions that follow the expected JSON schema. In the case of raw text, we ask the model to provide its output of each field by writing the field first, then its value.

**Prompt further details.** In total, we get $5 \times 2^4 = 80$ different prompts. We test each of them by making sure that a judge based on this prompt and using llama3.1-8B is able to recognize the correct output between an obviously bad and good completion when judging the pair with both orders.

Overall, our search space contains 4480 possible judge configurations, which corresponds to 80 different prompts and 56 choices for the 7 LLM models, 4 temperatures and the choice whether to average or not output permutations. We give more details in the appendix on the different prompting mechanisms as well as output preference formats.

## 5.3. Multi-fidelity and multi-objective optimization

**Multi-fidelity optimization.** Next, we perform multi-fidelity multi-objective optimization in order to find judge configurations that are good for both accuracy and cost while keeping the cost of the search feasible with multi-fidelity.

Evaluating all judges on all battles is expensive and would cost $\mathcal{N} \times \mathcal{P}$ annotations if we have $\mathcal{N}$ judges and $\mathcal{P}$ battles. One way to reduce the cost of the search is to apply a multifidelity approach such as sucessful-halving (Karnin et al., 2013).

We perform the tuning by running configurations in three steps. In the first step, we run all $\mathcal{N}$ configurations with only $\mathcal{P}/9$ battles. We then run the top $\mathcal{N}/3$ judges on $\mathcal{P}/3$ battles in a second step and finally run $\mathcal{N}/9$ on the full $\mathcal{P}$

---

**Prompt Template**

You are a highly efficient assistant, please evaluate and select the best large language model based on the quality of their responses to a given instruction.

**User Prompt:** Who is Geoffrey Hinton?
**Assistant A:** Geoffrey Hinton is a research scientist.
**Assistant B:** I do not know who Geoffrey Hinton is.

**# Your Output**
**## Format Description**
Your output should follow this format:
```
{
  "answer":  <your answer to the user
prompt>,
  "explanation":  <your explanation on
why you think A or B is better>,
  "score_A":  <between 0 and 10 to
rate the quality of A>,
  "score_B":  <between 0 and 10 to
rate the quality of B>
}
```
**## Your output, do not repeat the input above.**

---

*Figure 3.* Illustration of the prompt templating approach. We parametrize the prompt with the following hyperparameters: Provide answer, Provide explanation, Provide example, use JSON, output preference format. Given each of the $2^4 \times 5 = 80$ prompt hyperparameter, we generate a prompt like this one.

battles.

This reduces the cost of the full search from $\mathcal{N} \times \mathcal{P}$ to $\mathcal{N} \times \mathcal{P}/3$. One could also use a more aggressive cutoff and save further in the computation but this would naturally increase the risk of missing good configurations.

**Multi-objective optimization.** When sorting the top judge configurations, we have two objectives to take into account since we would like to find judges that both accurate and cheap. For instance, prompting a judge to perform chain-of-thought of to provide its answer may improve performance, but the extra-cost may be better spent on a better and more expensive base model.

To sort configurations while accounting for the two objectives, we use non-dominated sort (Emmerich & Deutz, 2018) as it was shown efficient in multi-fidelity settings (Salinas et al., 2021; Schmucker et al., 2021; Izquierdo et al., 2021). We illustrate the priority given to the judge configurations in Fig. 4 in the first and second selection step where one can see that the priority model the geometry of the Pareto front well.

## 5.4. Hyperparameter analysis

On Fig. 5, we show the validation performance of all judge configurations at the lowest fidelity. We see that while the number of parameters influence the performance, the prompt and other judge hyperparameters have a significant importance given the wide variation of performance obtained for a fixed model.

Next, we examine which hyperparameters and prompt characteristics contribute to the best judge performance. In Fig. 6, we analyze all 4480 judge configurations at the lowest fidelity (i.e., evaluated on 400 instructions) where we conduct a survival analysis, measuring how often each hyperparameter appears in the top 100 configurations with the highest human agreement. We perform this analysis separately for large models (blue bars) and smaller models (orange bars) to identify which hyperparameters are most effective in each case.

Without surprise, the model used for the LLM judge is the most impacting hyperparameters and larger is generally better. Llama3 performs better than Qwen2.5 and Gemma when used as a judge.

This analysis also reveals less obvious hindsights:

- The output format used to obtain judge preferences plays a big role, almost as important as the choice of the model used for the judge. Small models are more sensitive which is expected as they struggle more with more complex output mechanism. For both small and large models, the pair format performs better.

- Increasing temperature negatively impacts performance

- Averaging the judgement after evaluating a pair of outputs in both orders gives a performance boost

- Providing an example helps the judge provided that a large model is used as smaller models gets confused by this extra information

- Asking the judge to provide an explanation, or its answer hurts performance

- Using JSON does not impact much the performance

## 5.5. Prompt stability

Next, we investigate how much the performance of a prompt varies between different model judges in Fig. 7, where we show the correlation between different judge models on how well they perform under different prompts. For each of the $m = 7$ models, we measure the average human agreement for all the different $n = 2^4 \times 5 = 80$ prompts and compute the covariance matrix $XX^T$ where $X \in \mathbb{R}^{m \times n}$ is the matrix

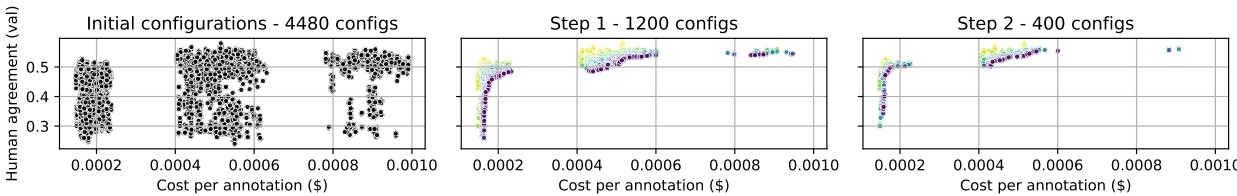

*Figure 4.* Illustration of the selection process. All 4480 configurations are first evaluated on 400 instructions (left), the top 1200 configurations are then evaluated on 1200 instructions (center) and finally the top 400 configurations are evaluated on 3548 instructions (right). The color denotes the ranking assigned by the non-dominated sort procedure.

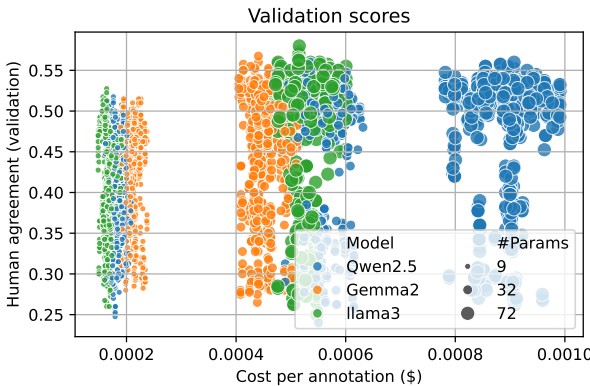

*Figure 5.* We plot the cost per annotation and human agreement of all 4480 judges when using 400 instructions. The model family and the number of parameters are represented with color and size respectively.

| Judge | Human agr. (↑) | Cost per 1K ann. (↓) |
|---|---|---|
| Random | 0.33 +/- 0.01 | - |
| Length | 0.42 +/- 0.01 | - |
| PandaLM-7B | 0.38 +/- 0.01 | 6.0 |
| JudgeLM-7B | 0.42 +/- 0.01 | 8.6 |
| Arena-Hard | 0.50 +/- 0.01 | 1.2 |
| Ours-small | 0.45 +/- 0.01 | 0.21 |
| Ours-medium | 0.47 +/- 0.01 | 0.48 |
| Ours-large | 0.49 +/- 0.01 | 0.48 |

*Table 2.* Comparison of judges on LMSys test instructions. For each judge, we report the bootstrap mean and std for human-agreement on 3K test instructions with 100 seeds.

of scores for all models and prompts. Interestingly, smaller models and larger models are highly correlated which shows that two group of prompts works well for large and small models. This is expected as lower capacity models may struggle to obey more complex instructions (e.g. rate models for style and accuracy using JSON) that are beneficial to evaluate better models.

### 5.6. Results on test datasets

In this section, we report the performance of three judges found by our search on several test sets. While the multi-objective search returns a list of judges with continuous cost tradeoffs as seen in Fig. 4, we report results for only 3 judges: small, medium and large with numbers of parameters respectively lower than 10B, lower than 32B and lower than 72B. We do this as it may offer an easier choice for a practitioner when picking a judge, for instance accounting for the memory constraint of a given GPU. For each category, we select the judge with the best validation score on the 3548 instructions of the validation set of LMSys.

| Judge | Human agr. (↑) |
|---|---|
| Random | 0.33 |
| Length | 0.60 |
| GPT-3.5 | 0.63 |
| GPT-4 | 0.67 |
| PandaLM-7B | 0.62 |
| PandaLM-70B | 0.67 |
| Ours-small | 0.67 |
| Ours-medium | 0.78 |
| Ours-large | 0.76 |

*Table 3.* Comparison with PandaLM on PandaLM test set. Note that our method is not fine-tuned as opposed to PandaLM.

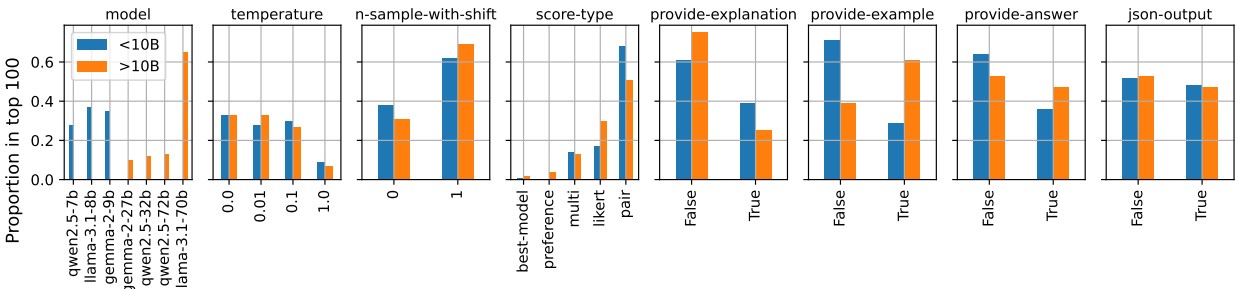

*Figure 6.* Fraction of time each hyperparameter appears in the top 100 configurations for small (<10B) and large models (>10B).

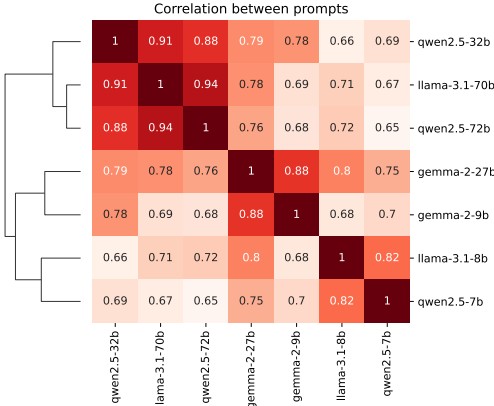

*Figure 7.* Prompt performance stability across different models. We show the correlation matrix between models when looking at their performance on all the 80 different prompts.

**LMSys.**  In Table 2, we compare tuned judges with a simple baseline that picks the longest answer, PandaLM (Wang et al., 2024), JudgeLM (Zhu et al., 2023) and Arena-Hard (Li et al., 2024). For the latter, we use GPT-4o mini to obtain a similar cost per annotation than other methods. We compute scores by measuring human-agreement on 3 000 test instructions that are not used for the model selection and report mean and standard errors over 100 bootstraps.

While all methods outperform a random baseline, Panda-LM-7B underperforms and JudgeLM-7B only matches a simple baseline *Length* that picks the longest answer. This is because the instructions on LMSys are more complex and significantly longer from the distribution used for fine-tuning and confirms previous findings that fine-tuned judges performance can be affected by change of distributions (Huang et al., 2024).

The judges we found outperforms all baselines and slightly underperforms or matches Arena-Hard for human-

| Judge | Sp. corr. (↑) | Cost per 1K ann. (↓) |
|---|---|---|
| Length | 0.50 +/- 0.21 | - |
| Arena-hard + Claude | 0.82 +/- 0.12 | 75.0 |
| Arena-hard + GPT4 | 0.90 +/- 0.06 | 50.0 |
| Ours-small | 0.81 +/- 0.10 | 0.21 |
| Ours-medium | 0.93 +/- 0.05 | 0.48 |
| Ours-large | 0.86 +/- 0.09 | 0.48 |

*Table 4.* For each judge, we compute the Spearman correlation between win-rates using the protocol of Arena-Hard and ELO-ratings computed from human annotations from Chatbot Arena. We report mean and std over 100 boostraps of the set of models.

agreement but strongly outperforms it in term of cost.

**PandaLM.**  In Table 3, we show the performance on PandaLM test set (Wang et al., 2024). The judges that we found all outperforms strongly PandaLM, even our small judge with less than 10B parameters outperforms its PandaLM counter with 70B parameters. Importantly, we recall that our approach only consider zeroshot judges, e.g. it does not fine on the training dataset which would improve further the performance on this dataset although it may also hurt generalization as seen for the LMSys dataset. We see that the 70B model underperforms slighly the 32B model however, both scores are high and close to inter-agreement rates seen in real-world data.

**Arena-Hard.**  In Table 4, we report the Spearman correlation of the judges found by our method, compared to the judge proposed in Arena-Hard using the 20 models available for both Claude-Opus and Gpt-4-1106-preview judges in the authors repository (Li et al., 2024). For each judge, we annotate all the 20 models on the 500 instructions of Arena-Hard against a baseline and compute winrates against this baseline. We then compute the Spearman correlation between the winrates and ELO-ratings from Chatbot Arena. We estimate the cost ofArena-Hard judges using the cost estimate from Ni et al. (2024) of 25$ to annotate one model on all 500 instructions and multiply this estimation by 50% for Claude

Opus given the difference as it corresponds roughly to the additional token cost compared to GPT-4-1106-preview.

The judge we found match or outperforms the judge considered at much lower cost. Importantly, the judge configurations are open-weight models which provide additional benefits, in particular for applications such as building a community leaderboard or building an open model.

## 6. Limitations

We currently select judges solely based on accuracy and cost, not on potential biases such as position (where LLMs favor responses based on order (Li et al., 2024)). To investigate whether our selection criteria worsen this bias, we measured the flip rate, how often a judge changes its decision when the response order is swapped, and found a strong negative correlation with the human agreement scores ($r = -0.789$) of the judge configurations of the top rung. While the position bias is not worsen, other biases, such as stylistic preferences or verbosity, may be worsen by our method. Future work could consider those biases by including them as additional objectives.

## 7. Conclusion

In this paper, we examined how judge performance is influenced by scaling and hyperparameter choices. We introduced a multi-fidelity, multi-objective approach to tune judge hyperparameters — including prompt design and base models — at a feasible cost. Our results demonstrate that this method can produce judges that outperform previous approaches across different budget constraints.

While some limitations of LLM judges persist, we hope that enabling cost-effective tuning will help the community refine their use and address remaining deficiencies. For example, our multi-objective procedure could be extended to optimize for additional criteria such as stability or explainability.

We release the code to reproduce our results, along with a dataset containing all annotations at `https://github.com/geoalgo/judgetuning`. We hope this resource will support further analysis and improvement of LLM judges.

## Impact Statement

This paper demonstrated how to tune judge hyperparameters while balancing cost considerations and ensuring that the search remains feasible. Our approach optimizes judge hyperparameters to maximize human agreement while also considering open-weight alternatives.

The benefits of this approach include enabling fairer and more cost-effective leaderboards and helping the community adopt judges that do not rely on closed systems. However, LLM judges may also reinforce undesirable superficial biases, such as favoring stylistic elements over substantive quality or perpetuating human biases present in the training data, including biases against certain groups. We conducted a preliminary review of the selected LMSys data and did not observe obvious issues, though our analysis was limited to a small sample. As a result, such systems should not be deployed without safeguards and additional bias evaluations.

Our approach analysed the prompting strategy and hyperparameters of the current generation of LLMs while we expect our conclusion to hold given the relative stability of prompt strategy across this family (see Fig. 7), the conclusion could change over time with the introduction of distinctive new capabilities such as reasoning.

## Acknowledgments

This research was partially supported by the following sources: TAILOR, a project funded by EU Horizon 2020 research and innovation programme under GA No 952215; the Deutsche Forschungsgemeinschaft (DFG, German Research Foundation) under grant number 417962828 and 539134284, through EFRE (FEIH_2698644) and the state of Baden-Württemberg; the European Research Council (ERC) Consolidator Grant "Deep Learning 2.0" (grant no. 101045765); EC under the grant No. 101195233 (OpenEuroLLM). Frank Hutter acknowledges financial support by the Hector Foundation. The authors acknowledge support from ELLIS and ELIZA. Views and opinions expressed are however those of the author(s) only and do not necessarily reflect those of the European Union or the ERC. Neither the European Union nor the ERC can be held responsible for them. The authors gratefully acknowledge the computing time made available to them on the high-performance computer NHR@KIT Compute Cluster at the NHR Center NHR@KIT. These Centers are jointly supported by the Federal Ministry of Education and Research and the state governments participating in the NHR (www.nhr-verein.de/unsere-partner).

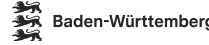 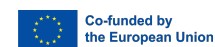 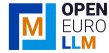

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

After analyzing the input prompt based on these criteria, you must list the criteria numbers that the prompt satisfies in the format of a Python array. For example, "Criteria Satisfied: [1, 2, 4, 6, 7]".

---

*Figure 8.* Prompt to evaluate instruction quality.

## A. Datasets

### A.1. Alpaca-Eval and Arena-Hard datasets

We consider two datasets that contains prompts, model completions and judge annotations for a grid of prompts and model pairs. The first one is Alpaca-Eval which contains 47 models completions on 805 prompts (Li et al., 2023). The second one is Arena-Hard which contains the completions on 500 instructions for 57 models (Li et al., 2024). In both cases, we select the 26 models that also appear in Chatbot Arena in order to be able to compute how well judge configurations approximate human judgement.

### A.2. LMSys

We use the LMSys dataset (lin Chiang et al., 2024) which contains 51734 battles and allows to measure the human-agreement of a given judge configuration.

**LMSys validation and test split.** When using LMSys, we rate instructions with the prompt from (Li et al., 2024) given in Fig. 8 and we use Llama3-8B-instruct which assigns a score to each instruction in [0, 7].

We select instructions which have a score greater than or equal to 5 and also have the criterion *1. Specificity* detected since it is important to avoid large ambiguity prompt to evaluate judge (e.g. we want to discard the instruction like "say hello" since they provide no value to distinguish models).

This gives 6548 instruction which we split randomly into 3548 validation instructions and 3000 test instructions. All model selection (e.g. non-dominated sort) is done only using validation instructions and only the best models on the validation set are evaluated on the test set.

## B. Experiment details

To generate inference with open models, we host the models locally using VLLM on L40 GPUs for models up to 32B parameters and on H100 GPUs for models with more than 70B parameters. Given that we use two clusters with two different job queues, we favored using *synchronous* successful halving rather than an asynchronous approach such as (Schmucker

| Hyperparameter | Values |
|---|---|
| model | Llama3 (8/70B), Qwen2.5 (7/27/70B), Gemma2 (9/27B) |
| temperature | 0.0, 0.01, 0.1, 1.0 |
| average-orders | False, True |
| provide-example | False, True |
| provide-answer | False, True |
| provide-explanation | False, True |
| use-json | False, True |
| output-type | likert, best-model-letter, pair, preference, multi |

*Table 5.* Judge Hyperparameters considered. In total, the search-space contains 4480 different judge configurations.

| Model | Cost / 1K token ($) |
|---|---|
| qwen2.5-72b | 0.58 |
| qwen2.5-32b | 0.36 |
| llama-3.1-70b | 0.35 |
| gemma-2-27b | 0.30 |
| gemma-2-9b | 0.14 |
| qwen2.5-7b | 0.12 |
| llama-3.1-8b | 0.11 |

*Table 6.* Cost per token estimated from our runtime evaluations

et al., 2021). We submitted all the 4480 configurations of the first fidelity with 400 instructions, then applied non-dominated sort and submitted the top 1200 configurations with 1200 instructions before finally submitting the top 400 configurations with 3548 instructions.

## B.1. Cost

To compute the cost of a judge annotation, we first estimate the token price and then multiply the number of tokens by the token price[3]. To obtain the average token price for a model, we measure the total runtime and number of tokens on a large collection of judge annotations for the given model. We then derive the cost per token using the H100 hourly price of runpod (2.79$/hour) for models requiring more than 48GB of VRAM (Llama3 70B, Qwen2.5 70B and Gemma2 27B) and using L40 hourly price for other models (0.99$/hour).

We arrive at the cost per token given in Table 6. The estimate are lower than a public provider such as Together which is expected as such a service needs to operate at a margin and over-provision machines to meet demand. The cost we estimate is highly conservative given that no optimization was done to optimize VLLM hyperparameters.

For close models, we compute the cost identically by multiplying tokens seen in prompt and completion with the corresponding token price.

## B.2. Prompt templating

**Prompt examples.** In Fig. 9, we show a full prompt corresponding to the prompt hyperparameter:

```
{ "Provide answer":  True, "Provide explanation":  True, "Provide example":
True, "use JSON":  True, "output preference format":  "Pair"}
```

and in Fig. 10, we show the prompt corresponding to:

```
{ "Provide answer":  True, "Provide explanation":  False, "Provide example":
False, "use JSON":  False, "output preference format":  "Likert"}.
```

---

[3]An alternative approach would be to measure cost by multiplying the runtime of a judge with the hourly price of the machine used. However, this approach yields to noisy estimation and large costs for some judges just because of hardware noise.

```
You are a highly efficient assistant, who evaluates and selects the best large language
↪ model based on the quality of their responses to a given instruction.
You will be shown one instruction and the output of Assistant A and Assistant B and will
↪ have to decide which one was best.
Make sure to not over-confidently prefer one assistant or the other and also make sure to
↪ not bias your preference based on the ordering or on the length of the answers.

# Example
Let us first look at one example.

## Input

<|User Prompt|>
What is the square root of 81? Just provide the answer.

<|The Start of Assistant A's Answer|>
The answer is 81, this can be seen as 9*9 = 81.
<|The End of Assistant A's Answer|>

<|The Start of Assistant B's Answer|>
81
<|The End of Assistant B's Answer|>
## Your expected output (must be a valid JSON)

```
{
  "answer": "81",
  "explanation": "Both model are correct however, the output from model A is verbose and
  ↪ does not provide just the answer whereas the instruction asked for conciseness.",
  "score_A": 2,
  "score_B": 8
}
```

For the explanation, do not exceed three sentences.

# Now is the judgement I would like you to make, please follow the format I just
↪ described.

## Input

<|User Prompt|>
Who is Barack Obama?

<|The Start of Assistant A's Answer|>
Barack Obama is a former US president.
<|The End of Assistant A's Answer|>

<|The Start of Assistant B's Answer|>
I do not know who Barack Obama is.
<|The End of Assistant B's Answer|>
## Your output, do not repeat the input above  (must be a valid JSON)
```
```

*Figure 9.* Example of a prompt for the user prompt "Who is Barack Obama?". In this case, the judge is asked to provide its answer, an explanation, and is provided an example. It is asked to use the Pair format and provide its answer in JSON. .

```
You are a highly efficient assistant, who evaluates and selects the best large language
↪  model based on the quality of their responses to a given instruction.
You will be shown one instruction and the output of Assistant A and Assistant B and will
↪  have to decide which one was best.
Make sure to not over-confidently prefer one assistant or the other and also make sure to
↪  not bias your preference based on the ordering or on the length of the answers.

<|User Prompt|>
Who is Barack Obama?

<|The Start of Assistant A's Answer|>
Barack Obama is a former US president.
<|The End of Assistant A's Answer|>

<|The Start of Assistant B's Answer|>
I do not know who Barack Obama is.
<|The End of Assistant B's Answer|>

# Your output

## Format description
Your output should follow this format:
```
answer: <your answer to the user prompt>
score: <one of A>>B, A>B, A=B, A<B, A<<B, see instruction bellow>
```
The "score" value should indicate your preference for the assistant. You must output only
↪  one of the following choices as your final verdict with a label:

A>>B: Assistant A is significantly better
A>B: Assistant A is slightly better
A=B: Tie, relatively the same
B>A: Assistant B is significantly better
B>>A: Assistant B is significantly better

## Your output, do not repeat the input above
```
```

*Figure 10.* Example of a prompt for the user prompt "Who is Barack Obama?". In this case, the judge is asked to provide its answer. It is asked to use the Likert format and provide its answer in raw text.

### B.3. Tuning cost estimation

**Alpaca-Eval and Arena-Hard.**  In both cases, to evaluate Spearman correlation one must annotate a judge on a grid of models and instructions.

Let us assume we evaluate $n_{\text{judges}} = 4480$ as done in this work for $n_{\text{models}} = 20$ models as done in (Li et al., 2024) on the 805 instructions of Alpaca Eval. We get the cost to annotated one model as 24\$ by using the estimation of Ni et al. (2024). This gives a cost of $4480 * 24 * 20 = 2\,186\,240\$$ for Alpaca-Eval and a cost of $2\,240\,000\$$ for Arena-Hard whose cost to annotate the instructions for one model was estimated to 25\$ in (Ni et al., 2024).

**Cost estimation of our approach.**  We evaluate $\mathcal{N} = 4480$ judge configurations on 400 instructions, then the top 1200 judge configurations on 1200 instructions, then the top 400 judge configurations on the full set of $\mathcal{P} = 3548$ validation instructions. This requires a total of $4\,651\,200$ annotations. On average, a single annotation takes about 0.6s on a H100. If using runpod with a cost of 2.79\$/hour per H100 hour, we get a total cost of $4651200/3600 * 0.6 * 2.79 \approx 2.1\text{K}\$$.

The savings are obtained by identifying a more efficient metrics to distinguish judges than Spearman correlation (which requires evaluating a grid of models and instructions in (Li et al., 2023) and (Li et al., 2024)) and applying multi-fidelity which allows us to save roughly a factor of 3 since it avoids to annotate the full set of $\mathcal{N} \times \mathcal{P}$.

## C. Multi-objective background

In hyperparameter-optimization, one seeks to find the best hyperparameter $\theta^*$ of a blackbox function $f : \mathbb{R}^d \to \mathbb{R}$, e.g. to find:

$$\theta^* = \arg\min_{\theta \in \mathbb{R}^d} f(\theta).$$

The blackbox may be for instance a neural network that we want to train and the hyperparameter may include the number of layers or the learning rate.

**Multi-objective optimization.**  When considering several objectives, we now want to minimize a function $f(\theta) \in \mathbb{R}^m$. Since we have more than one objective, there is not a single best hyperparameter $\theta^*$ in general but a set of non-dominated solutions.

We say that a hyperparameter $\theta$ dominates $\theta'$ if and only if:

$$\forall i, f(\theta)_i \leq f(\theta')_i \quad \text{and} \quad \exists i, f(\theta)_i < f(\theta')_i$$

and denotes it with $\theta \prec \theta'$ i.e. when all components of $f(\theta)$ are lower or equal than the ones of $f(\theta')$ and one component is strictly better.

We aim to find the Pareto front $\mathcal{P}$ which consists of non-dominated solution:

$$\mathcal{P} = \{\theta \in \mathbb{R}^d \mid \; \nexists \theta', \theta' \prec \theta\}$$

**Non-dominated sort.**  When applying successful-halving, we need to sort the top configurations, for instance to keep the top 50% and let those configurations run with a larger budget.

Since we have multiple objectives, something must be done to adapt the algorithm. While averaging out the objective (or doing more advanced scalarization) allows to go back to the scalar case, it only works on some cases (where the Pareto front is convex in the case of averaging the objectives for instance).

Another approach is to use non-dominated sort (Emmerich & Deutz, 2018) which we illustrate in Fig. 11 and now describe. The approach first computes the Pareto front of the current set of observations and assign top-ranks with a heuristic to break ties. Then, the approach is applied recursively until no points are left. To break ties, multiple heuristic can be used, in our case we use an epsilon-net as it was shown to perform well in (Schmucker et al., 2021). As can be seen in Fig. 4, the approach models the geometry of the Pareto front compared to scalarization approaches.

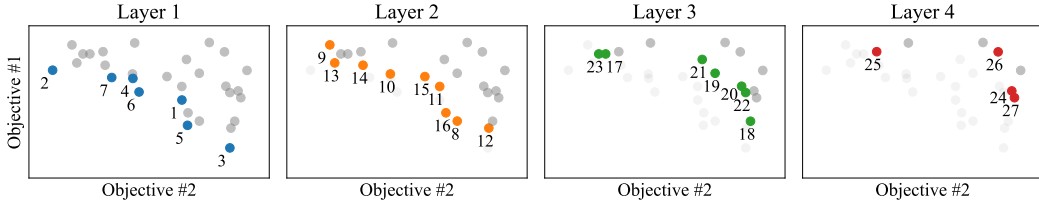

*Figure 11.* Illustration of the non-dominated sorting approach. The process first computes the Pareto front, assigning top ranks to the points in this layer. Next, the Pareto front is determined for the remaining points, which are then assigned the next set of rankings. This process continues iteratively until all points are ranked. To resolve ties within each layer, a heuristic is applied to balance both sparsity and coverage.

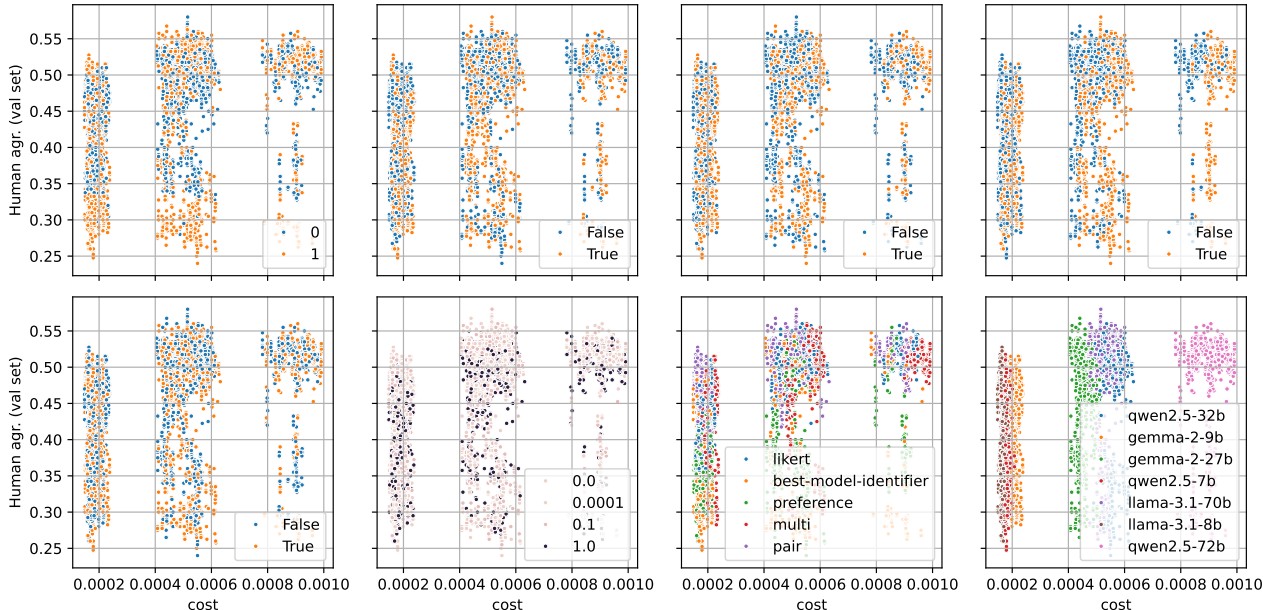

*Figure 12.* Scatter plot of cost and human agreement on the 400 validation instructions for all judges. We color-code each hyperparameter differently to illustrate the performance of all judges. Even using the same LLM model, there is a large spread of performance when varying other hyperparameters without an obvious pattern to distinguish the best configuration which motivates the need to search for optimal hyperparameters.

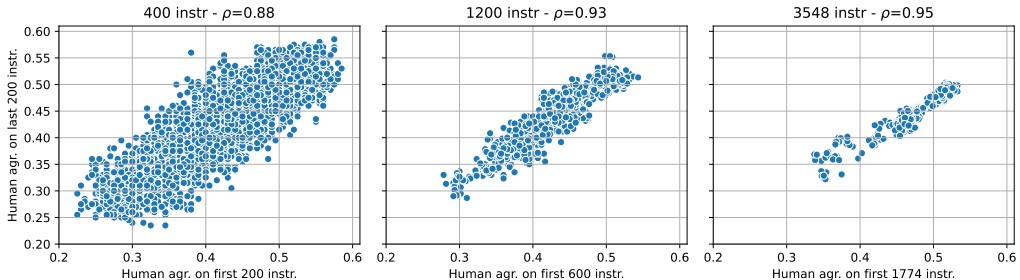

*Figure 13.* Correlation for different fidelity sizes. For each fidelity, we randomly split the instructions into two buckets and plot the human-agreement on the first bucket versus the same metric computed on the second bucket of instructions for all the available judges, we also report the Spearman correlation $\rho$ between the two groups. This allows to see the correlation one can obtain between the two sets. For the final fidelity, we measure the validation performance on 3548 instructions and use 3000 test instructions so we expect a higher correlation between validation and test scores.

