# OpenReview forum: "Tuning LLM Judge Design Decisions for 1/1000 of the Cost"
_ICML.cc/2025/Conference — ICML 2025 poster_

### Official Review · Reviewer_tVtX · 2025-03-08

**Overall Recommendation:** 4

**Summary:**

This paper aims to broadly profile the various factors impacting judge LLM performance and results, systematically analyzing the impact of factors related to prompt, hyperparameter selection, answer extraction method, and model design. It adopts a cost-effective approach to minimize search overhead while preserving performance. The paper identifies how to improve over previous SOTA baselines, presenting configurations which notably boost judge LLM performance in relevant downstream settings.

---
**Update after rebuttal**

Thanks for the authors’ response, I have reviewed the rebuttal and discussions and will be keeping my score.

**Claims And Evidence:**

Claims are generally supported in the paper.

**Essential References Not Discussed:**

N/A — To my understanding the paper sufficiently covers relevant key works for the LLM-as-a-Judge paradigm.

**Experimental Designs Or Analyses:**

Experiments utilize a range of model families and sizes, with sensible selections of inference hyperparameters. Analyses are relevant and insightful across all settings.

**Methods And Evaluation Criteria:**

The paper develops a comprehensive evaluation framework, considering a diverse array of prompting strategies in addition to having good coverage of variations related to model selection, to profile the impact of various decisions involved in using LLMs as judges.

**Other Comments Or Suggestions:**

Minor formatting suggestions
* Bullet points of key contributions in Section 1 utilize no period for the first three bullets versus use of a period in the fourth bullet.

**Other Strengths And Weaknesses:**

Strengths
* The paper exhibits good organization and writing quality. Figures are well-presented and easy to read and understand.
* The paper answers an important need in the community toward understanding and improving use of LLMs as judges in various settings.

Weaknesses
* Experiments could consider finer gradations of inference temperature.
* Confidence elicitation in LLMs is generally dependent on prompt wording and the range of confidence scores the model is asked to produce. How did the paper elicit confidence and were variations in elicitation strategy considered?

**Questions For Authors:**

N/A

**Relation To Broader Scientific Literature:**

The paper provides a complementary set of findings to existing work concerning judge LLMs, systematically characterizing the impact of model selection and other experimental factors on judge LLM performance. Recommendations provided in the paper serve to guide design choices in future research involving judge LLMs.

**Theoretical Claims:**

The theory presented in the paper is sound.

---

> ### Author Rebuttal · Authors · 2025-03-28
>
> Thank you for your review and for reviewing all appendices. We are delighted that you found the analyses relevant and insightful across all settings.
>
> Please find our answer for the two points raised in your review.
>
> **Experiments could consider finer gradations of inference temperature.**
>
> Indeed you are right that the optimal temperature may not be exactly covered with the grid we selected. Our intent was to analyse the range of temperatures that works to guide practitioners when considering judges. We think that just getting the range may be enough for the temperature to get good performance (e.g. a low temperature is enough to get most of the performance; at least it is not the dominant hyperparameter to select as opposed to the base model or the prompting strategy).
>
> **Confidence elicitation in LLMs is generally dependent on prompt wording and the range of confidence scores the model is asked to produce. How did the paper elicit confidence and were variations in elicitation strategy considered?**
>
> We totally agree, the range is indeed important, and LLM judges performance indeed differ when their performance is graded between [1-5] or [0-10] for instance. For confidence, we elicited it by asking for a score in [0, 1]. We agree that setting a score in [1-5] for instance may change the performance with respect to this hyperparameter. We tried other ranges in smaller scale experiments but we did not observe a significant effect of this hyperparameter on performance so we decided on a simple option.
>
> Here is the prompting corresponding to confidence, we will add it to the appendix.
> Your output should follow this format:
> ```
> answer: <your answer to the user prompt>
> confidence: <a number between 0 and 1 to indicate your confidence>
> best_assistant: <either "A" or "B" to indicate which Assistant gave the best answer>
> ```

---

### Official Review · Reviewer_8kjw · 2025-03-10

**Overall Recommendation:** 3

**Summary:**

This paper proposes an efficient way to finetune LLM judges via a multi-objective multi-fidelity approach.

**Claims And Evidence:**

The claim is supported by experiments on three models and three datasets. Therefore the result is convincing.

**Essential References Not Discussed:**

No missing reference found.

**Experimental Designs Or Analyses:**

No significant problems in experiments or analyses.

**Methods And Evaluation Criteria:**

No significant flaws in method and evaluation.

**Other Comments Or Suggestions:**

The number 1/1000 in the title does not have clear support in the main text. Therefore, it should be replaced with non-quantitative words like "lower".

**Other Strengths And Weaknesses:**

No additional strength or weakness.

**Questions For Authors:**

1. How does the prompt template look like when the prompt hyperparameters include "provide confidence" (an option included in Page 5) ?
2. Can you provide the resulting prompt template after judge tuning in one of your experiment setting?

**Relation To Broader Scientific Literature:**

This paper pushes forward the research in LLM judge tuning.

**Theoretical Claims:**

Not applicable for this paper.

---

> ### Author Rebuttal · Authors · 2025-03-28
>
> Thank you for your review. We are delighted to hear that you found the results convincing.
>
> Please find our answers below to the three points raised in your review.
>
> **The number 1/1000 in the title does not have clear support in the main text. Therefore, it should be replaced with non-quantitative words like "lower".**
>
> In the second page of the main text, line 106 we made this statement:
> “We estimate that our approach costs approximately \\$2k to search through 4 480 judge configurations and that evaluating the same number of judges using Alpaca-Eval or Arena-Hard methodology would cost around \$2M, see B.3 for details.”
> This is where the 1/1000 numbers came. We will avoid the footnote to make it easier to find.
>
> **How does the prompt template look like when the prompt hyperparameters include "provide confidence" (an option included in Page 5) ?**
>
> The prompt to ask for confidence looks like this, thank you for the callout, we added this point in the appendix as we realized that this prompt for this hyperparameter was missing.
> Your output should follow this format:
>
> ```
> answer: <your answer to the user prompt>
> confidence: <a number between 0 and 1 to indicate your confidence>
> best_assistant: <either "A" or "B" to indicate which Assistant gave the best answer>
> ```
>
> **Can you provide the resulting prompt template after judge tuning in one of your experiment setting?**
>
> Of-course, here is the prompt found for the middle size LLM. It is a great suggestion we will add it to the appendix as it is revealing for a reader.
>
> ```
> You are a highly efficient assistant, who evaluates and selects the best large language model based on the quality of their responses to a given instruction.
> You will be shown one instruction and the output of Assistant A and Assistant B and will have to decide which one was best.
> Make sure to not over-confidently prefer one assistant or the other and also make sure to not bias your preference based on the ordering or on the length of the answers.
>
> <|User Prompt|>
> {USER_PROMPT}
>
> <|The Start of Assistant A's Answer|>
> {ANSWER_A}
> <|The End of Assistant A's Answer|>
>
> <|The Start of Assistant B's Answer|>
> {ANSWER_B}
> <|The End of Assistant B's Answer|>
>
> # Your output
>
> ## Format description
> Your output should follow this format:
> \```
> answer: <your answer to the user prompt>
> score_A: <a number between 0 and 10 to indicate the quality of Assistant A's answer>
> score_B: <a number between 0 and 10 to indicate the quality of Assistant B's answer>
> \```
>
> ## Your output, do not repeat the input above
> ```

---

### Official Review · Reviewer_rLGG · 2025-03-13

**Overall Recommendation:** 3

**Summary:**

This paper is more like an extensive experimental report. The authors systematically analyze the hyperparameters of LLM judges, including the choice of model, inference parameters, and prompt hyperparameters (e.g., output format, provide answer or other information, JSON formatting). Overall, our search space contains 4480 possible judge configurations, which corresponds to 80 different prompts and 56 choices for the 7 LLM models, 4 temperatures, and the choice of whether to average or not output permutations. They leverage multi-objective and multi-fidelity optimization techniques to balance accuracy and cost, significantly reducing the expense of tuning compared to prior methods. By evaluating on datasets including Alpaca-Eval, Arena-Hard, and LMSys, the optimized judges outperform existing methods in terms of accuracy and cost-efficiency while relying solely on open-weight models. The study highlights the importance of prompt design and model selection in improving judge performance and provides insights into the trade-offs between cost and accuracy. The results show that their approach can identify superior judge configurations at a fraction of the cost, making the evaluation of LLMs more accessible and efficient.

**Claims And Evidence:**

The claims made in the paper are generally well-supported by clear and convincing evidence. The authors provide an approach to optimizing LLM judges through extensive experimentation and analysis.

**Essential References Not Discussed:**

I think there are several related works that need further discussion, especially the works on automatic prompt optimization. It seems to me that the authors completely ignore the works related to this relevant topic.
1. Fairer Preferences Elicit Improved Human-Aligned Large Language Model Judgments (EMNLP 2024): a prompt optimizer for bridging the gap between LLM evaluators and human judgments.
2. Aligning with Human Judgement: The Role of Pairwise Preference in Large Language Model Evaluators (COLM 2024): an uncertainty-guided search-based rank aggregation method for LLM judges.

**Experimental Designs Or Analyses:**

I think most of the experimental designs of this paper are sound. I only have one question about the effectiveness of multi-objective optimization. The authors claim that their multi-objective and multi-fidelity optimization approach significantly reduces the cost of tuning LLM judges while maintaining or improving performance. However, the paper could benefit from a more detailed comparison with other optimization techniques to further validate the superiority of their approach.

**Methods And Evaluation Criteria:**

In general, the proposed method and evaluation criteria make sense for the problem.

**Other Comments Or Suggestions:**

No

**Other Strengths And Weaknesses:**

My primary concern is that the authors claim that their optimized judges are cost-effective and accessible due to the use of open-weight models. However, they do not address the potential challenges of maintaining performance over time as new LLMs or datasets emerge. The adaptability of their optimized judges to future changes in LLM technology or evaluation standards could be a potential area of concern. Do the insights summarized above (prompt design, decoding temperatures, etc.) still hold for later LLMs, such as reasoning LLMs such as OpenAI o1 or QwQ (DeepSeek R1 is released on Jan 2025, it is OK that this paper does not test it)?

It seems to me that the methodology appears to rely heavily on brute-force search across configurations without presenting a more principled approach to parameter optimization. The paper does not adequately address whether the identified optimal configurations generalize or are merely artifacts of the specific LLMs being evaluated, also this paper ignored the related works on automatic prompt optimization (see the first paper in the section of "Essential References Not Discussed").

**Questions For Authors:**

See the Other Strengths And Weaknesses above.

**Relation To Broader Scientific Literature:**

The key contributions of the paper are closely related to the broader scientific literature on LLM-as-Judge and prompt engineering.
Their analysis also reveals several insights of using LLM judges:
1. Without surprise, the model used for the LLM judge is the most impacting hyperparameters and larger is generally better.
2. The output format used to obtain judge preferences plays a big role.
3. Increasing temperature negatively impacts performance.
4. Averaging the judgement after evaluating a pair of outputs in both orders gives a performance boost.
5. Providing an example helps the judge provided that a large model is used as smaller models gets confused by this extra information.
6. Asking the judge to provide an explanation, or its answer hurts performance.
7. Using JSON does not impact much the performance.

**Theoretical Claims:**

This paper does not include any theoretical claims. It is more like an experimental report.

---

> ### Author Rebuttal · Authors · 2025-03-28
>
> Thank you for your review and valuable feedback. We answer here the main three points raised by your review.
>
> **Comparison with other search approaches. The methodology relies on brute-force search.**
>
> We believe the characterization of brute-force may be a bit strong since we are using an efficient multi-fidelity approach but we agree with you that we evaluate a full grid of all options. Of-course, an alternative could be to use a model-based approach which can sample a better part of the space, for instance a Gaussian Process or an Evolution Algorithm. However, we prefer to explore the whole search space for two important reasons. The first one is that the analysis of hyperparameter performance is easier and does not require fitting for instance a random-forest to understand hyperparameter importance. The second is that having the full set of options allows one to simulate any (model-based) method and compare their performance. We will make the data available in a way where one can easily compare different optimizers, we did not do it in this paper as we believe this is a point orthogonal to the main point of the paper (showing how LLM judges can be tuned by reducing the tuning cost). Note that the cost of tuning on judge configuration with our approach is still reasonable costing around \$285 per judge.
>
> **This paper ignored the related works on automatic prompt optimization.**
>
> Thank you for providing these relevant references. We cited two papers related to prompt optimization (Fernando et al. 2023 and Doddapaneni et al. 2024), but we agree that both references are relevant, and will discuss them in the paper. The first one in particular is a great reference as it is another one discussing prompt tuning for judges. Regarding the second one, it is relevant for LLM judges in general but does not mention prompt tuning. Is it the correct one that you wanted to discuss?
>
> **The paper does not adequately address whether the identified optimal configurations generalize or are merely artifacts of the specific LLMs being evaluated.**
>
> You are right that different LLMs may behave differently regarding the prompting strategy. We did one analysis regarding this with Figure 7, which shows how much prompt ranking changes across different models. The figure showed that similar model capacities have high correlation (the Spearman correlation of qwen2.5-32b, llama-3.1-70b, and qwen2.5-72 is larger than 0.88 for instance).
>
> We agree that this analysis does not fully resolve this question, but we believe it quantifies how much we can expect our results to generalize through current LLMs (also it is clear that no absolute statement can be made for LLMs as future generations may always differ from the current one in any aspect). We also want to point out that our method costs about 2k dollars to evaluate all the 4480 configurations for the 7 families which gives a cost of 285 dollars per LLM, which we believe is reasonable for a new model.
>
> Following your remark, we propose to add the following statement in the Impact Statement:  “Our approach analysed the prompting strategy and hyperparameters of the current generation of LLMs while we expect our conclusion to hold given the relative stability of prompt strategy across this family (see Fig 7), the conclusion could change over time with the introduction of distinctive new capabilities such as reasoning.”
>
> Let us know if you have any feedback on the wording or generally regarding this point.

---

> > ### Comment · Reviewer_rLGG · 2025-04-04
> >
> > Thanks for the clarifications; they have addressed some of my concerns. I will increase my rating to weak acceptance for this paper. For missing references, the second one is not related to automated prompt optimization (sorry for the expression); it is an uncertainty-guided search-based rank aggregation method for LLM judges that might be worth noting.

---

> > > ### Author Response · Authors · 2025-04-04
> > >
> > > Thank you for the update. The second paper is indeed quite relevant for LLM judges and we will make sure to reference it.

---

### Official Review · Reviewer_ETKe · 2025-03-15

**Overall Recommendation:** 4

**Summary:**

This paper proposes a cost-effective approach to systematically tune hyperparameters of Large Language Model (LLM)-based judges for evaluating other LLMs, significantly reducing the required resources. The authors leverage a multi-objective, multi-fidelity optimization framework to efficiently search through 4,480 configurations, considering factors like model choice, prompting strategy, inference parameters, and output parsing method. Their method identifies judges outperforming existing benchmarks in accuracy and cost while exclusively utilizing open-weight models to ensure accessibility and reproducibility. The authors find that optimal judge performance strongly depends on the selected LLM, prompt style, temperature settings, and response parsing mechanism, rather than simply scaling model size or instruction count.

## update after rebuttal
I thank the authors for the rebuttal, but I keep my score.

**Claims And Evidence:**

The claims made in the paper are supported by clear and convincing evidence, primarily through systematic empirical experiments comparing the tuned judges against existing benchmarks across multiple datasets. The extensive hyperparameter search, multi-objective optimization analysis, and direct comparisons against baselines strongly substantiate their conclusions.

**Essential References Not Discussed:**

None

**Experimental Designs Or Analyses:**

The multi-fidelity procedure (successive halving across three fidelity steps: 400, 1200, and 3548 instructions) is logically sound and efficiently manages computational costs. The analyses on hyperparameter sensitivity (e.g., prompt formatting, model size, and temperature) are well-executed, clearly demonstrating which factors contribute significantly to judge performance. There were no issues identified; the approach is robust, transparent, and methodologically sound.

**Methods And Evaluation Criteria:**

The methods and evaluation criteria proposed by the authors make sense for the problem they’re tackling. They use human agreement and correlations with established datasets (LMSys, PandaLM, Arena-Hard) as evaluation metrics, which fits nicely with their goal. Their choice to optimize hyperparameters using a cost-saving, step-wise tuning method is logical given the high expense typically involved in evaluating these models.

**Other Comments Or Suggestions:**

None

**Other Strengths And Weaknesses:**

Strengths:
- The use of open-weight, zero-shot models increases accessibility and encourages reproducibility, addressing critical limitations of prior work relying on expensive closed models.
- The analysis of hyperparameter importance (prompt style, inference parameters) is thorough and insightful, providing valuable practical guidelines.

Weaknesses:
- Potential biases or superficial stylistic preferences of judges, despite being briefly mentioned, are not systematically analyzed in depth.

**Questions For Authors:**

You briefly mention potential superficial biases (e.g., preference for longer answers). Could you provide further insights or data on how your tuned judges mitigate or exacerbate such biases compared to existing approaches?

**Relation To Broader Scientific Literature:**

The paper positions itself within existing research on automatic evaluation of LLMs using LLM judges, emphasizing the cost and complexity of human annotation as motivation. The authors specifically build upon previous work like Alpaca-Eval and Arena-Hard, systematically addressing confounding factors (such as simultaneous changes in judge model, prompts, and scoring methods) that hindered clear comparisons in prior research. By employing multi-objective multi-fidelity optimization, they extend beyond existing prompt-tuning approaches (e.g., Promptbreeder) and earlier explorations into prompt stability across models. Unlike related efforts such as PandaLM and JudgeLM—which primarily use fine-tuning or closed models—their approach notably relies exclusively on open-weight, zero-shot judges, significantly enhancing accessibility and reproducibility.

**Theoretical Claims:**

NA

---

> ### Author Rebuttal · Authors · 2025-03-28
>
> Thank you for your feedback and thorough review. We are delighted to hear that you found the approach sound and well-executed.
>
> Your point on potential bias is indeed very relevant. One concern one could have is that the selection could on one hand improve human agreement but, on the other hand worsen bias. We analyzed the position bias and found that it was negatively correlated with human-agreement performance, e.g. better models for human-agreement tend to have lower positional bias. For length, we could not conduct an analysis in time.
>
> We will add a discussion of this point in our paper. In particular, we will point that in case improving human-agreement worsens bias (which could be the case for length for instance), one could then just add this measured bias as an objective with our proposed approach.

---

> > ### Comment · Reviewer_ETKe · 2025-04-02
> >
> > Thank you for your response. It would indeed be a valuable point to add.

---

### Decision · Program_Chairs · 2025-05-01

**Decision:**

Accept (poster)

**Comment:**

This paper proposes a cost-effective approach to systematically tune hyperparameters of LLM-as-a-judge for evaluating other LLMs, significantly reducing the required resources. The authors leverage a multi-objective, multi-fidelity optimization framework to efficiently search through 4,480 configurations, considering factors like model choice, prompting strategy, inference parameters, and output parsing method. By evaluating on datasets including Alpaca-Eval, Arena-Hard, and LMSys, the optimized judges outperform existing methods in terms of accuracy and cost-efficiency while relying solely on open-weight models.

The paper answers an important need in the community toward understanding and improving use of LLMs as judges in various settings. The use of open-weight, zero-shot models increases accessibility and encourages reproducibility, addressing critical limitations of prior work relying on expensive closed models. The analysis of hyperparameter importance (prompt style, inference parameters) is thorough and insightful, providing valuable practical guidelines.

Please add the missing references about prompt optimization in your related work section.